# The “Hockey” Assist Makes the Difference—Validation of a Defensive Disruptiveness Model to Evaluate Passing Sequences in Elite Soccer

**DOI:** 10.3390/e23121607

**Published:** 2021-11-30

**Authors:** Leander Forcher, Matthias Kempe, Stefan Altmann, Leon Forcher, Alexander Woll

**Affiliations:** 1Institute of Sport and Sport Science (IfSS), Karlsruher Institute of Technology (KIT), 76131 Karlsruhe, Germany; Stefan.altmann@kit.edu (S.A.); Leon.forcher@kit.edu (L.F.); Alexander.woll@kit.edu (A.W.); 2Center for Human Movement Sciences, University of Groningen, University Medical Center Groningen (UMCG), 9713 GZ Groningen, The Netherlands; M.kempe@umcg.nl; 3TSG ResearchLab gGmbH, 74939 Zuzenhausen, Germany; 4TSG 1899 Hoffenheim, 74939 Zuzenhausen, Germany

**Keywords:** soccer, performance indicators, passing, tracking data, spatiotemporal data, sports analytics

## Abstract

With the growing availability of position data in sports, spatiotemporal analysis in soccer is a topic of rising interest. The aim of this study is to validate a performance indicator, namely D-Def, measuring passing effectiveness. D-Def calculates the change of the teams’ centroid, centroids of formation lines (e.g., defensive line), teams’ surface area, and teams’ spread in the following three seconds after a pass and therefore results in a measure of disruption of the opponents’ defense following a pass. While this measure was introduced earlier, in this study we aim to prove the usefulness to evaluate attacking sequences. In this study, 258 games of Dutch Eredivisie season 2018/19 were included, resulting in 13,094 attacks. D-Def, pass length, pass velocity, and pass angle of the last four passes of each attack were calculated and compared between successful and unsuccessful attacks. D-Def showed higher values for passes of successful compared to unsuccessful attacks (0.001 < *p* ≤ 0.029, 0.06 ≤ d ≤ 0.23). This difference showed the highest effects sizes in the penultimate pass (d = 0.23) and the maximal D-Def value of an attack (d = 0.23). Passing length (0.001 < *p* ≤ 0.236, 0.08 ≤ d ≤ 0.17) and passing velocity (0.001 < *p* ≤ 0.690, −0.09 ≤ d ≤ 0.12) showed inconsistent results in discriminating between successful and unsuccessful attacks. The results indicate that D-Def is a useful indicator for the measurement of pass effectiveness in attacking sequences, highlighting that successful attacks are connected to disruptive passing. Within successful attacks, at least one high disruptive action (pass with D-Def > 28) needs to be present. In addition, the penultimate pass (“hockey assist”) of an attack seems crucial in characterizing successful attacks.

## 1. Introduction

With the ongoing digitalization in sports, the importance and usage of data in soccer has grown constantly in the last decades [1,2]. In line with this development, the use of spatiotemporal tracking data for tactical game analysis is seen as a beneficial tool to produce more objective, time-efficient, and in-depth analyses [3,4]. In the past, match analysis focused on the physical and technical performance of soccer players [5,6]. With this development of new analysis methods using tracking data, the tactical performance is analyzed more frequently, for example using measures of collective organization (e.g., centroids, spread measures) [7]. By the analysis of tracking data, the exact positioning of players, their spatial formation, and the inter-player distance can be measured more easily and objective [8]. With it, the dynamic, interactive, and complex nature of soccer can be considered to better characterize performance in soccer, which was unattended in traditional (e.g., notational) analysis approaches [2].

In this context, the investigation of passing behavior received broad attention as it is the most frequently occurring tactical pattern in a game [9,10] and therefore is one of the key skills in soccer [11]. In addition, effective passes are seen as crucial for successful performance [12] and team success in soccer [13].

In the past, the analysis of passing behavior usually did not go beyond numbers of passes and various forms of completion rates [11] revealing little information about passing performance and overlooking factors such as value or reward of passes. However, with the stated availability of spatiotemporal tracking data more complex investigations and key performance metrics on passing behavior in soccer have been developed in recent years [9,10,12,14].

On one hand, there have been several investigations from a computational view that resulted in various prediction models or the estimation of the risk and reward of a pass [9,10]. On the other hand, there have been investigations from sport science research groups to measure passing effectiveness resulting in key performance metrics such as the effect of passes on outplayed opponents and space control in the final third [12] or *Dangerousity* (a metric quantifying the danger of any attacking action or a whole attack to end in a goal) [14].

While these approaches and performance indicators provide new insights and translate practical ideas into numbers, they mostly also overvalue forward passes as they solely connect effective passing to scoring opportunities and advancing in close proximity of the goal [15]. In doing so, these approaches overvalue specific passes (e.g., assists) that might be crucial for the actual development of a successful attack. To account for the complex dynamics of build-ups and space creation in all areas of the pitch, not only the spaces in front of the goal, Goes et al. [15] introduced a passing evaluation tool called D-Def. D-Def is based on the idea that an effective pass disrupts the opposing defense and creates space to achieve scoring opportunities. In this context, other studies showed a higher chance of scoring against an unbalanced defense compared to a balanced defense [16,17]. Earlier studies already revealed that D-Def can differentiate between good and bad passers on the individual level [15] and is a better predictor of the final result of a match compared to other performance indicators [18]. However, since the end result of a game is strongly influenced by chance [19], the relationship of D-Def and success should not solely be traced back to the end result. Consequently, in this study, we investigate the connection of D-Def with the success of a team in the context of individual attacks.

The evaluation of possessions and attacking sequences and therefore also the evaluation of key performance indicators in soccer is generally based on summation scores or singular events. Following this approach, the majority of published spatiotemporal analysis studies using tracking data reduce their analysis on one single summarized value for a whole game or a possession (e.g., total number of shots on goal, percentage of passing accuracy) (Lepschy, 2021; Dufour, 2017). Other studies compare individual possessions with each other using summarized values [20] or focus on the valuation of single actions like shots on goal (e.g., expected goals) [21,22] or passes [15]. In doing so, they do not reveal evidence about the consecutive chain of actions in the attacking process. The approach of the analysis of consecutive actions is strengthened by Sarmento et al. [2] who pointed out that the analysis of sequential aspects of the game is important to increase the practical impact of match analysis in soccer. Using this approach, it cannot be discovered which pass in an attack was actually important or if some actions benefit from each other (e.g., dribblings are possibly more effective after switching sides with a long pass across the pitch). Given the complex and partially chaotic structure of soccer, it is important to get insights into the chain of events in an attack (e.g., which action of an attack is most important for success?) to further validate performance indicators, as the actual significance of them is still pending [23].

While the idea of analyzing chains of actions is already established in other sports, like tennis or American football [24,25] (e.g., using Markov chains) it is seldomly used in soccer.

One study using this approach was conducted by Kempe and Memmert [26] investigating consecutive actions in the sequence of an attack to evaluate the influence of creative action on goal scoring. In this study, the authors rated the creativity of the last eight actions of attacks that led to goals in the Football FIFA World Cup 2010 and 2014, as well as the Football UEFA Euro 2016. While this study provided empirical evidence for the importance of creativity in soccer, the authors used a rather simple study design with notational data and only focused on goal scoring attacks.

In the present study, the goal is to establish a new approach for the analysis of attacks by investigating individual consecutive actions (passes) in the chain of an attack using the opportunities of tracking data.

Therefore, the purposes of this study are to use this approach (investigation of consecutive actions in the chain of an attack) (i) to test whether the quantitative pass model, D-Def, is a valid measure for pass effectiveness on a single possession level and (ii) to reveal practice-relevant information about the characteristics of successful passing in the attacking process in elite soccer. To do so, we will differentiate between passes of successful and unsuccessful attacks using a spatiotemporal tool that objectively measures the danger of an attack. Accounting for the first aim, we hypothesized that passes of successful attacks show higher D-Def values compared to unsuccessful attacks. For the second aim, an explanatory approach of the consecutive actions of an attack will be applied.

## 2. Materials and Methods

### 2.1. Data

We used an observational design in which position tracking data and event data of 258 games of the Dutch Eredivisie season 2018/19 were included. Of all 305 games of the whole season, 47 matches (15.4%) were excluded because of erroneous or missing data.

The position data of all 22 players and the ball as well as the associated ball event data was collected by a semi-automatic optical tracking system (TRACAB, Chyron Hego, Melville, NY, USA) that measures the X- and Y-coordinates of all players and the ball with a sampling frequency of 25 Hz. This camera-based tracking system has recently been validated [27]. Before the data processing and analysis, the raw position data were pre-processed on a match-by-match basis with ImoClient software (Inmotiotec Object Tracking B.V., The Netherlands). Pre-processing included filtering with a weighted Gaussian algorithm (100% sensitivity), downsampling to 10 Hz, and automatic detection of possession and ball events based on synchronization of position tracking data with tagged event data.

Both tracking data and event data were imported in Python 3.8.3 and data processing and data analyses were conducted using the NumPy, Pandas, Math, SciPy, and Matplotlib libraries.

### 2.2. Success of Attacks

Every ball possession (attack) of each team was detected using the ball event data. A ball possession started with a team gaining control over the ball and ended whenever the opponent gained control over the ball again or there was a stoppage of play (ball out of bounds, free kick, corner, goal kick, or goal). To investigate passing behavior, the focus was on deliberate attacks only, as they include several deliberate (intentional) passes in a row. Therefore, only attacks that lasted longer than five seconds and had a minimum of three passes were selected.

The success of an attack was quantified with a danger value based on the tracking data. It was calculated with a zone value from which points were deducted depending on the defensive pressure of the opposing team. The zone value was measured using a grid similar to Link’s work Dangerousity [14] which represents the last 35 m in front of the opposing goal, showing higher values the closer the player gets to the opposing goal. The amount of defensive pressure was operationalized through the positions (regarding the position of the goal) and distances of the defenders in close proximity to the ball leading player.

For every passing reception, the danger value (ranging between 0 and 1) was calculated. Accordingly, attacks with a peak danger value >0 were classified as successful and attacks with a peak danger value of 0 were classified as unsuccessful [28]. Hence, successful attacks represent a player in control over the ball, in at least 30 m range to the opposing goal with insufficient defensive pressure and the potential of creating a scoring opportunity.

### 2.3. Passing Effectiveness

Passing effectiveness was measured using an overall composite measure for the disruptiveness of the defensive organization following a pass, D-Def [15]. It is based on the idea that effective passes disrupt the opponents’ defensive organization and create space to achieve scoring opportunities.

To measure defensive organization on the pitch, D-Def uses three different components: centroids (of full team, defensive-line, midfielder-line, and attacking-line), surface area of full team, and spread of full team. To measure the centroids of the formation lines, the tactical formations were automatically determined using a K-Means clustering (n_clusters = 3) algorithm of the players’ average position in the first half. For every team, two superior formations, one attacking formation and one defending formation, were specified in three lines (e.g., 4-4-2, 4-3-3, 3-5-2). Accordingly, the players were assigned to the three lines.

The centroids of the full team (C_x_, C_Y_), of defensive-line (C_xdef_, C_ydef_), of midfielder-line (C_xmid_, C_ymid_), and of attacking-line (C_xatt_, C_yatt_) were calculated by the average position in X- and Y-coordinates separately (Equations (1) and (2)). The surface area (S_surface_) was calculated as the smallest convex hull area of the positions of all players at a given timestamp t (Equation (3)). The spread (S_spread_) was calculated by the Frobenius norm of the positions of all players at a given timestamp t (Equation (3)). D-Def calculates the change of these three measures between the moment a pass was played and three seconds following this pass. According to the results of the principal component analysis in the work of Goes et al. [15], D-Def consists of three components (PC1, PC2, and PC3) (Equation (4)), which result in the measure of D-Def with a range from 0 to 150 (with 0 indicating no disruption and 150 indicating a maximum of disruption).
(1)$$\mathrm{PC}1=-0.46\;C_{x}+0.26\;C_{y}-0.43\;C_{\mathrm{xdef}}+0.24\;C_{\mathrm{ydef}}-0.43\;C_{\mathrm{xmid}}+0.24\;C_{\mathrm{ymid}}-0.41\;C_{\mathrm{xatt}}+0.24\;C_{\mathrm{yatt}}$$
(2)$$\mathrm{PC}2=-0.26\;C_{x}-0.47\;C_{y}-0.24\;C_{\mathrm{xdef}}-0.43\;C_{\mathrm{ydef}}-0.25\;C_{\mathrm{xmid}}-0.43\;C_{\mathrm{ymid}}-0.24\;C_{\mathrm{xatt}}\;\;-0.40\;C_{\mathrm{yatt}}$$
(3)$$\mathrm{PC}3=0.71\;S_{\mathrm{area}}\;+0.71\;S_{\mathrm{spread}}$$
(4)D-Def=|PC1|+|PC2|+|PC3|

For every attack that was considered, the last four passes were investigated. For every pass, the effectiveness (D-Def), the passing length (L_pass_ in [m]) ((Equation (5)), passing velocity (V_pass_ in [m/s]) ((Equation (6)), and passing angle (α in [°]) were calculated (Equations (7) and (8)) (see Figure 1).
(5)$$L_{\mathrm{pass}}=\surd \left({\left(X_{\mathrm{reception}}\;-\;X_{\mathrm{pass}}\right)}^{2}+{\left(Y_{\mathrm{reception}}\;-\;Y_{\mathrm{pass}}\right)}^{2}\right)$$
(6)$$V_{\mathrm{pass}}=L_{\mathrm{pass}}\;÷\;(t_{\mathrm{reception}}\;-\;t_{\mathrm{pass}})\;$$
(7)$$m_{\mathrm{pass}}=(Y_{\mathrm{reception}}\;-\;Y_{\mathrm{pass}})\;÷\;(X_{\mathrm{reception}}\;-\;X_{\mathrm{pass}})$$
(8)$$\alpha =\tan^{-1}(m_{\mathrm{pass}})$$

m_pass_ = slope of passing vector

α = pass angle

L_pass_ = pass length

V_pass_ = pass velocity

Y_reception_ = Y-coordinate of reception

X_reception_ = X-coordinate of reception

Y_pass_ = Y-coordinate of pass

X_pass_ = Y-coordinate of pass

t_reception_ = time of reception

t_pass_ = time of pass

### 2.4. Statistical Analyses

To test for the hypothesis that D-Def varies in successful and unsuccessful attacks, a two-way ANOVA (2 × 4) with repeated measures was conducted with repeated measure being the passing sequence (last 4 passes of an attack: pass 1 (last pass), pass 2 (penultimate pass), pass 3 (third last pass), and pass 4 (fourth last pass)) and success of an attack (groups: successful attacks vs unsuccessful attacks). If Mauchly’s test of sphericity was significant the Greenhouse–Geisser correction was used.

To investigate the differences between successful and unsuccessful attacks in more detail and to account for the explanatory part of this study, independent *t*-tests were used to compare means of D-Def, pass length, and pass velocity of pass 1 (last pass), pass 2 (penultimate pass), pass 3 (third last pass), pass 4 (fourth last pass), average and maximum of an attack between successful and unsuccessful attacks. To avoid alpha-error, the Bonferroni–Holm correction was used. Before testing, Levene’s test for homogeneity of variances was conducted for every comparison and if significant the correction was used.

To compare passing angles, circular statistics were used. A Watson–Williams F-test was conducted between successful and unsuccessful attacks for angles of every pass (pass 1, pass 2, pass 3, pass 4) separately. This test examines whether a set of mean directions are equal [29].

Additionally, effect sizes were calculated. In variance analyses partial eta squared (η^2^) was determined as effect size with η^2^ < 0.06 representing a small effect, 0.06 ≤ η^2^ ≤ 0.14 representing a medium effect, and η^2^ > 0.14 representing a large effect [30]. For *t*-tests, Cohen’s d was calculated as effect size with d < 0.5 representing a small effect, 0.5 ≤ d ≤ 0.8 representing a medium effect, and d > 0.8 representing a large effect [31].

All statistical analyses were conducted using IBM SPSS Statistics 25.0.0.0 (IBM Co., New York, NY, USA). The level of significance was set to *p* < 0.05.

## 3. Results

All 258 full matches included in this study resulted in 13,094 considered attacks. From all considered attacks, 7565 attacks were classified as unsuccessful (57.8%) and 5529 attacks were classified as successful (42.2%) with a mean *danger value* of 0.27 ± 0.26. Successful attacks lasted significantly longer (successful: 40.41 ± 74.42 [s]; unsuccessful: 29.20 ± 71.00 [s]) and had significantly more passes (successful: 7.14 ± 4.39; unsuccessful: 5.26 ± 2.92) compared to unsuccessful attacks (*p* < 0.001).

The results of all ANOVAs are depicted in Table 1 and the results of pairwise comparisons and descriptive statistics between successful and unsuccessful attacks are shown in Table 2.

The ANOVA with D-Def as independent variable showed significant results between the passes in the attacking sequence (*p* < 0.001), between successful and unsuccessful attacks, (*p* < 0.001), and the interaction (*p* < 0.001). While the interaction showed trivial effect size (η^2^ < 0.01) and the success effect showed medium effect size (η^2^ = 0.14), the passing sequence effect showed large effect size (η^2^ = 0.23). The pairwise comparisons of the individual passes (*t*-tests), depicted in Figure 2, indicated that D-Def values of all passes of successful attacks were significantly higher (0.001 < *p* ≤ 0.029) compared to unsuccessful attacks. Furthermore, the mean and maximum D-Def value of an attack was significantly higher in successful attacks compared to unsuccessful attacks (*p* < 0.001) (see Figure 2). This difference showed the highest effect size in the penultimate pass (d = 0.23) and the maximum of an attack (d = 0.23).

The ANOVAs with pass length and pass velocity as independent variable revealed similar results. The passing sequence, the success of an attack, and the interaction showed significant results in both variables (*p* < 0.001). The difference between successful and unsuccessful attacks as well as the interaction showed trivial effect sizes (η^2^ ≤ 0.01) for both passing length and velocity. The differences between the consecutive passes of an attack showed a medium effect size for passing length (η^2^ = 0.12) and a large effect size for passing velocity (η^2^ = 0.18). In the pairwise comparisons of individual passes between successful and unsuccessful attacks, pass length and pass velocity showed similar but inconsistent results. Pass 2 of successful attacks showed significantly longer and faster passes compared to unsuccessful attacks (*p* < 0.001). Pass 3 and 4 showed no significant differences in passing length and velocity. Pass 1 of successful attacks showed smaller passing velocity compared to unsuccessful attacks (*p* = 0.002).

The passing angles of the last four passes of successful attacks differed significantly from the passing angles of associated passes of unsuccessful attacks (Pass 1: *p* < 0.001; Pass 2: *p* < 0.001; Pass 3: *p* < 0.001; Pass 4: *p* < 0.001) (see Figure 3).

## 4. Discussion

The purpose of this study was to investigate passing in soccer using a seldomly used approach by investigating the consecutive chain of actions in the attacking process. Using this approach, we aimed (i) to show the validity of the pass effectiveness model, D-Def, and (ii) to reveal practice-relevant information about successful passing in elite soccer. To account for the first aim, we showed significant differences in D-Def of passes between successful and unsuccessful attacks. For the second purpose, we focus on the characteristics of passes of successful attacks and their differences to passes of unsuccessful attacks.

In general, the reported percentage distribution between successful (42.2%) and unsuccessful (57.8%) attacks showed fewer differences compared to studies using similar approaches. For example, Goes et al. [28] observed less than 10% successful attacks. This variation can be traced back to the less-strict inclusion criteria of a minimum of three consecutive passes to filter deliberate attacks compared to Goes et al. [28] who used the minimum of 5 s combined with a set starting location (in the first third) of an attack. The present approach increased the sample benefits and therefore, the statistical power and presentiveness of our analysis. Furthermore, the results of general attacking attributes (duration and number of passes) are comparable to results of investigations in other leagues [32]. Hence, the following results can be interpreted with a higher generality.

Assessing the effectiveness of passing, we used D-Def as a compositive measure of defensive disruptiveness following a pass. D-Def values consistently showed significant higher values in passes of successful attacks compared to passes of unsuccessful attacks (see Table 2), supporting the hypothesis that successful attacks are associated with passes with higher D-Def values. This result supports the idea that D-Def can be used to evaluated passing performance. This is in line with earlier studies demonstrating that D-Def can differentiate between good and bad passers on an individual level [15], and D-Def being a better predictor of the end result of a game in comparison to other key performance indicators [18]. This connection of D-Def with the end result can now be expanded with the current finding of D-Def showing higher values in successful attacks. This indicates the connection of D-Def to success on a single possession level. Hence, it can be concluded, that D-Def is a valid measurement of passing effectiveness.

Furthermore, the analysis of consecutive actions in the chain of an attack reveals that this difference of D-Def values between successful and unsuccessful attacks is the biggest in the penultimate pass and the maximum of an attack. It can be derived, that one pass of an attack has to show a high D-Def value (D-Def > 28) to result in a successful possession and that the penultimate pass (also called “hockey assist”) is a crucial moment of an attack in order to create time and space for the receiver to assist a scoring opportunity. This importance of the penultimate action (“hockey assist”) of an attack can be supported by the findings of Kempe et al. [26] who investigated the creativity of consecutive actions that led to goals. They discovered that the in goal scoring attacks the last three actions (hockey assist, assist, and shot on goal) of an attack are more creative than the previous ones and the creativity is increasing towards the end of an attack [26]. Comparably, a steady increase of D-Def values towards the last actions of an attack, with the last pass showing the highest D-Def values, was found in this study. Both results show that towards the end of an attack the increase of effectiveness of actions (creativity or passing effectiveness) is important for the success of an attack.

Subsequently, the results of the passing attributes (length, velocity, and angle) are discussed to answer the second research question of this paper about the characteristics of successful passing. The differences of passing length and passing velocity between successful and unsuccessful attacks showed trivial effect sizes in the ANOVAs and inconsistent results in *t*-tests. Those findings indicate that these two passing attributes are not highly decisive for successful passing. Furthermore, there are more prominent differences between the consecutive passes of an attack (medium effect size for passing length & large effect size for passing velocity in ANOVAs), for example, the last pass of an attack is strikingly longer and faster than the previous passes, compared to the differences between successful and unsuccessful attacks. In contrast, passing angles show significant differences between passes of successful and unsuccessful attacks. Overall, the proportions of passing angles reveal that few passes are played vertically forwards and backwards. In unsuccessful attacks, the distribution shows more sideways passes, and in successful attacks, more diagonal passes forward as well as backward. This outcome points out, that forward passes are not played very frequently and successful passing is not clearly connected to forward passing (as passes of successful attacks do not show more forward passes compared to passes of unsuccessful attacks). This strengthens the idea of the current approach that effective passing is not necessarily connected to forward passing which was a main criticism on other approaches evaluating passing behavior [10,12,33].

Summarizing those findings, passes of successful attacks are often played diagonally (forwards and backwards), with a medium length (Ø = 18.8 (m)), a medium velocity (Ø = 11.7 (m/s)) and a high D-Def value (Ø = 27.7) indicating high disruption of the opposing defense. Those findings are comparable to earlier studies that discovered that effective passes are between 19–30 (m) long and with a passing velocity of 10.7 (m/s) [15]. However, the results of passing length and passing velocity should not be generalized as our results indicate that they are not highly decisive for successful passing. In contrast, for the success of an attack it is more important that on one side passes should not be played horizontally too frequently and on the other side, passes should show a high disruption of the opposing defense (high D-Def values). This disruption of the defense is mostly decisive as it creates space with fewer temporal and spatial pressure for receiving players and therefore can entail scoring opportunities in dangerous areas of the pitch. Those practice-relevant information about successful passing can help practitioners to create training regimes, analyze opponents, or rate players (e.g., for player recruitment), etc. [12]. Furthermore, the practice-oriented analysis of consecutive actions of an attack used in this study can help practitioners to transfer research findings to practice [2]. For example, the finding of the importance of the penultimate action of an attack can be implemented in training and match analyses.

There are some limitations of this study that should be noted. Firstly, the strong inclusion criteria to filter deliberate attacks compared to other approaches [28] result in a lot of short attacks that were excluded. Secondly, the high variance of the results due to the large sample size used should be noted in the interpretation of effect sizes. Lastly, in this investigation, only the last four passes of an attack were considered, and effects of previous passes are not depicted. However, both the findings of Kempe and Memmert [26] and this study indicate the importance of actions towards the end of an attack.

However, this investigation showed several strengths compared to similar studies. At first, the advanced study design makes it possible to reveal insights about the effectiveness of individual actions in the chain of an attack. Furthermore, the specific calculation of the investigated performance indicator D-Def takes the complex interactions with the opponent into account and therefore portrays the complex interaction of two soccer teams [34]. Additionally, by combining a large data set collected in professional competition (one full season of Dutch Eredivisie) with the methodology of D-Def that enables capturing complex patterns, conclusions can be made with a stronger ecological validity [4].

## 5. Conclusions

This study is an example of a sport science work that utilizes observational designs in which large data sets were collected in competition and used for the validation of a feature to assess some aspect of performance [3,12,14], in this case, D-Def [15]. We found that passes of successful attacks have significantly higher D-Def values compared to passes of unsuccessful attacks and thereby demonstrated the validity of this construct. Furthermore, we illustrated the constitutes of successful passing behavior to reveal practice-relevant information. Future research should focus on the collaboration between sport science and computer science, which will gain further importance to introduce the necessary expertise into the sport science domain [23] and will provide paths of new methods and key performance metrics into applied settings. This evolution of match analysis has the potential to change how soccer is analyzed and will define how a team can be successful in the future.

## Figures and Tables

**Figure 1 entropy-23-01607-f001:**
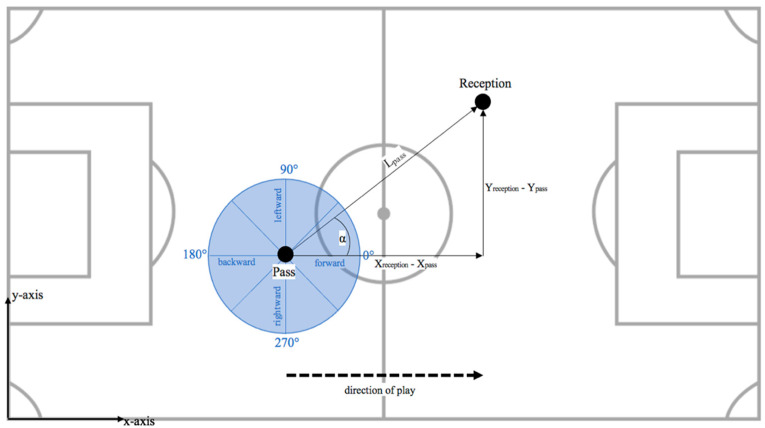
Graphical representation of passing length (Lpass) with X- (Xreception—Xpass) and Y-direction (Yreception—Ypass) as well as the passing angle (α) of an exemplary pass from player 1 (Pass) to player 2 (Reception).

**Figure 2 entropy-23-01607-f002:**
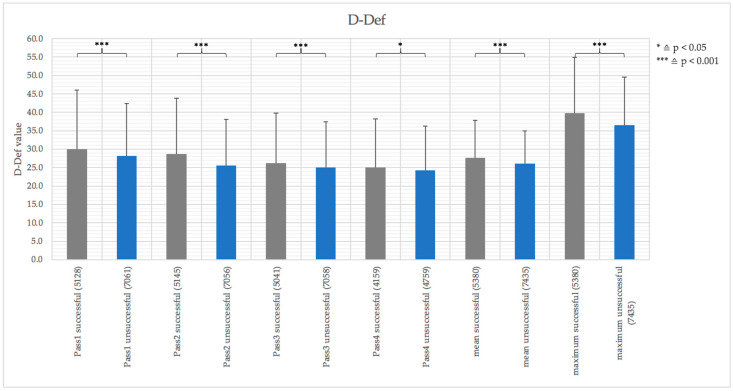
Means and their standard deviation of D-Def values of the last four passes, as well as mean and maximum of these 4 last passes, separated in successful attacks (black) and unsuccessful attacks (blue) with associated sample sizes in brackets.

**Figure 3 entropy-23-01607-f003:**
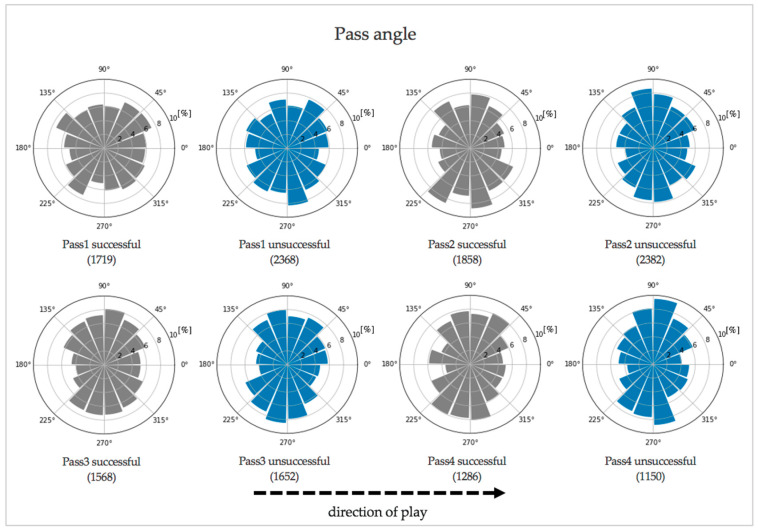
Percentage distribution of the passing angles of the last four passes of an attack grouped in successful (grey) and unsuccessful (blue) attacks with associated sample sizes in brackets.

**Table 1 entropy-23-01607-t001:** Results of ANOVAs of D-Def, passing length, and passing velocity (* ≙ *p* < 0.05).

	df	Mean Square	F	*p*	η ^2^
D-Def					
Success of attacks (successful, unsuccessful attacks)	1	35,266.95	110.31	<0.001 *	0.14
Pass sequence (pass1, pass2, pass3, pass4)	3	24,909.81	185.04	<0.001 *	0.23
Success of attacks * Pass sequence	3	989.04	16.76	<0.001 *	<0.01
error (success of attacks)	8000	319.71			
Pass length					
Success of attacks (successful, unsuccessful attacks)	1	3658.50	22.47	<0.001 *	0.01
Pass sequence (pass1, pass2, pass3, pass4)	3	38,500.50	306.52	<0.001 *	0.12
Success of attacks * Pass sequence	3	852.80	6.79	<0.001 *	<0.01
error (success of attacks)	2258	162.85			
Pass velocity					
Success of attacks (successful, unsuccessful attacks)	1	604.96	16.38	<0.001 *	<0.01
Pass sequence (pass1, pass2, pass3, pass4)	3	14,718.74	497.91	<0.001 *	0.18
Success of attacks * Pass sequence	3	553.54	18.73	<0.001 *	<0.01
error (success of attacks)	2218	36.92			

**Table 2 entropy-23-01607-t002:** Results of *t*-tests between successful and unsuccessful attacks. Independent variables are depicted with D-Def values, passing length, and passing velocity of the last four passes of an attack as well as the mean and maximum of an attack.

	Successful Attacks			Unsuccessful Attacks			Independent *t*-Test			
	N	Mean	SD	N	Mean	SD	df	T	*p*	d
duration of attack (s)	5529	40.41	74.42	7565	29.20	71.00	13,092	8.75	<0.001	0.16
passes of attack	5529	7.14	4.39	7565	5.26	2.92	13,092	29.41	<0.001	0.52
D-Def pass 1	5128	29.96	16.02	7061	28.16	14.20	12,187	6.55	<0.001	0.12
D-Def pass 2	5145	28.69	15.13	7056	25.51	12.63	12,199	12.66	<0.001	0.23
D-Def pass 3	5041	26.26	13.56	7058	24.99	12.43	12,097	5.33	<0.001	0.10
D-Def pass 4	4159	25.01	13.21	4759	24.25	12.02	8916	2.87	0.029	0.06
D-Def mean	5380	27.67	10.10	7435	26.01	9.00	12,813	9.76	<0.001	0.18
D-Def max	5380	39.77	15.10	7435	36.54	13.03	12,813	12.95	<0.001	0.23
pass length pass 1 (m)	3441	23.97	15.93	4589	22.75	15.36	8028	3.44	0.005	0.08
pass length pass 2 (m)	3737	17.32	12.86	4691	16.19	11.48	8426	4.23	<0.001	0.09
pass length pass 3 (m)	3105	16.62	11.83	3259	16.28	11.03	6362	1.18	0.681	
pass length pass 4 (m)	2552	16.09	11.03	2238	16.47	10.82	4788	−1.21	0.681	
pass length mean (m)	5079	18.80	10.90	6573	17.74	10.71	11,650	5.26	<0.001	0.10
pass length max (m)	5079	26.86	15.55	6573	24.31	14.90	11,650	8.99	<0.001	0.17
pass velocity pass 1 (m/s)	3202	15.55	9.48	4192	16.41	10.37	7392	−3.66	0.002	−0.09
pass velocity pass 2 (m/s)	3711	10.50	4.49	4660	9.93	4.82	8369	5.58	<0.001	0.12
pass velocity pass 3 (m/s)	3085	10.37	4.61	3233	10.13	4.71	6316	2.05	0.245	
pass velocity pass 4 (m/s)	2541	10.12	4.39	2222	9.94	4.27	4761	1.44	0.600	
pass velocity mean (m/s)	4992	11.67	5.28	6362	11.87	6.50	11,352	−1.74	0.408	
pass velocity max (m/s)	4992	15.36	8.13	6362	15.42	9.14	11,352	−0.40	0.689	

## Data Availability

Data examples are available upon request to the contact author.

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
