# Peer review of "The “Hockey” Assist Makes the Difference—Validation of a Defensive Disruptiveness Model to Evaluate Passing Sequences in Elite Soccer"

_entropy, 2021, doi:10.3390/e23121607_

Round 1

Reviewer 1 Report

Having read the article " The “hockey” assist makes the difference - Validation of a defensive disruptiveness model to evaluate passing sequences in elite soccer", I believe there can be an interesting contribution for the current state of the art in this specific topic. The clarity and flow in some parts still need improvement. Hence, I've recommended revision to improve further text clarity before I can consider recommending it for publishing, according the following reasons:

The introduction section synthetizes the current state of the art. Nerveless, some major and actual references in this context are not discussed (e.g. the systematic review of Sarmento et al (2018), Sports medicine, What performance analysts… - See also the reviews of Clemente et al…) I would like to see a deeper presentation of the current state of the art and the influence that a broader range of factors can have in this process.

The methods are described in detailed way. Well done!

Results -

Pag 6 – line 236 – please reformulate: “Therof 7565…”

The authors should reflect more deeply about their results and should provide useful information for coaches and sport scientists. Although the discussion is well structured, a more comprehensive overview of the usefulness of this type of analysis should be discussed according to the possible implications from training implementation.

Additionally, a limitations section is missing.

Please change “game analysis” by “match analysis” in conclusion. There are important differences between the concepts and specialized literature tend to adopted “match analysis”.

Author Response

Dear Dr. Clemente,

Dear Reviewers,

The authors would like to thank the editor and the reviewers for their careful consideration and constructive criticism of this manuscript. We appreciate their positive comments, at the same time, agree with their suggestions for change, and have now revised our manuscript accordingly. Please find below a detailed point-by-point response.

The Authors

Having read the article " The “hockey” assist makes the difference - Validation of a defensive disruptiveness model to evaluate passing sequences in elite soccer", I believe there can be an interesting contribution for the current state of the art in this specific topic. The clarity and flow in some parts still need improvement. Hence, I've recommended revision to improve further text clarity before I can consider recommending it for publishing, according the following reasons:

A: Thank you for your constructive and helpful feedback. We tried to address each of your comments adequately (please see below).

The introduction section synthetizes the current state of the art. Nerveless, some major and actual references in this context are not discussed (e.g. the systematic review of Sarmento et al (2018), Sports medicine, What performance analysts… - See also the reviews of Clemente et al…) I would like to see a deeper presentation of the current state of the art and the influence that a broader range of factors can have in this process.

A: Thank you for raising this important point. According to your comment, we made several changes to the introduction section. We considered the mentioned references and discussed their findings according to the topic of our manuscript. With it, we presented the current state of the art in match analysis more deeply.

Please see lines 41-52 : „ In the past, match analysis focused on physical and technical performance of soccer players [5,6]. With this development of new analysis methods using tracking data, the tactical performance is analyzed more frequently, for example using measures of collective organization (e.g. centroids, spread measures) [7]. By the analysis of tracking data, the exact positioning of players, their spatial formation, and the inter-player distance can be measured more easily and objective [8]. With it, the dynamic, interactive and complex nature of soccer can be considered to better characterize performance in soccer, which was unattended in traditional (e.g. notational) analysis approaches [2].“

And lines 94-97: “The approach of the analysis of consecutive actions is strengthened by Sarmento et al. [2]who pointed out that the analysis of sequential aspects of the game is important to increase practical impact of match analysis in soccer.“

The methods are described in detailed way. Well done!

A: Thank you very much for the positive feedback.

Results - Page 6 – line 236 – please reformulate: “Therof 7565…”

A: We rewrote the mentioned sentence and hope that it is more suitable now.

Please see line 253-255: “From all considered attacks, 7565 attacks were classified as unsuccessful (57.8 %) and 5529 attacks were classified as successful (42.2 %) with a mean danger value of 0.27 ± 0.26

The authors should reflect more deeply about their results and should provide useful information for coaches and sport scientists. Although the discussion is well structured, a more comprehensive overview of the usefulness of this type of analysis should be discussed according to the possible implications from training implementation.

A: Thank you for making us aware of the importance of clarifying the practical use of the findings of this manuscript. We agree that the information that is useful for coaches and sport scientists is not clearly stated. Accordingly we added further information in the discussion section and addressed the practical use of the findings.

Please see lines 374-385: “In contrast, for the success of an attack it is more important that on one side passes should not be played horizontally too frequently and on the other side, passes should show a high disruption of the opposing defense (high D-Def values). This disruption of the defense is mostly decisive as it creates space with fewer temporal and spatial pressure for receiving players and therefore can entail scoring opportunities in dangerous areas of the pitch. Those practice-relevant information about successful passing can help practitioners to create training regimes, analyze opponents, or rate players (e.g. for player recruitment), etc. [12]. Furthermore, the practice-oriented analysis of consecutive actions of an attack used in this study can help practitioners to transfer research findings to practice [2]. For example, the finding of the importance of the penultimate action of an attack can be implemented in training and match analyses.“

Additionally, a limitations section is missing.

A: A limitation section was part of the original manuscript. If the stated limitations should be edited or extended, we welcome you to point this out to us. We discuss the limitations of our study in lines 386-392: “There are some limitations of this study that should be noted. Firstly, the strong inclusion criteria to filter deliberate attacks compared to other approaches [29] result in a lot of short attacks that were excluded. Secondly, the high variance of the results due to the large sample size used should be noted in the interpretation of effect sizes. Lastly, in this investigation, only the last four passes of an attack were considered, and effects of previous passes are not depicted. However, both the findings of Kempe and Memmert [27] and this study indicate the importance of actions towards the end of an attack.

Please change “game analysis” by “match analysis” in conclusion. There are important differences between the concepts and specialized literature tend to adopted “match analysis”.

A: We agree with your recommendation and changed the wording to match analysis. See line 412.

Reviewer 2 Report

The manuscript examines whether D-Def is a useful parameter to distinguish between successful and unsuccessful passes. The manuscript is very well written, easy to follow for a wide-ranging audience and of interest.

Abstract

It would be useful in the abstract to add 1 additional sentence explaining a little more about D-Def, for those who are unfamiliar with it. What do you mean by passing effectiveness?

Introduction

The introduction is excellent and provides a nice overview of the research to date, whilst providing good justification for the study.

Methods

Line 120. Provide the validity data for TRACAB.

Results

Figure 2. Only include positive y-error bars.

Discussion

An excellent discussion which provides very good explanations for the findings as well as outlining the limitations of the research.

Author Response

Dear Dr. Clemente,

Dear Reviewers,

The authors would like to thank the editor and the reviewers for their careful consideration and constructive criticism of this manuscript. We appreciate their positive comments, at the same time, agree with their suggestions for change, and have now revised our manuscript accordingly. Please find below a detailed point-by-point response.

The Authors

The manuscript examines whether D-Def is a useful parameter to distinguish between successful and unsuccessful passes. The manuscript is very well written, easy to follow for a wide-ranging audience and of interest. 

A: Thank you for your constructive and helpful feedback. We tried to address each of your comments adequately (please see below).

Abstract

It would be useful in the abstract to add 1 additional sentence explaining a little more about D-Def, for those who are unfamiliar with it. What do you mean by passing effectiveness?

A: Thank you for your recommendation. We agree that there has to be an additional explanation about D-Def to make the abstract more clear for those who are unfamiliar with D-Def. We added an additional sentence in the abstract. Please see line 18-21: “D-Def calculates the change of the teams’ centroid, centroids of formation lines (e.g. defensive line), teams’ surface area, and teams’ spread in the following three seconds after a pass and therefore results in a measure of disruption of the opponents’ defense following a pass.”

Introduction

The introduction is excellent and provides a nice overview of the research to date, whilst providing good justification for the study.

A: Thank you very much for the positive feedback.

Methods

Line 120. Provide the validity data for TRACAB.

A: According to you comment we added an additional sentence in the method section and provided the citation of the associated study. Please see lines 134-135: “This camera-based tracking system has recently been validated [28].”

Results

Figure 2. Only include positive y-error bars.

A: Thank you for your comment on Figure 2. Accordingly, we changed Figure 2 and removed the negative y-error bars. Please see line 276.

Discussion

An excellent discussion which provides very good explanations for the findings as well as outlining the limitations of the research.

A: Thank you very much for the positive feedback.

Round 2

Reviewer 1 Report

I would like to congratulate the authors by this revised version of the paper.